# Feasibility Analysis of Biogas Production by Using GIS and Multicriteria Decision Aid Methods in the Central African Republic

Francis Auguste Fleury Junior Dima [1,2,*], Zifu Li [1,2,*], Heinz-Peter Mang [1] and Lixin Zhu [2]

1   School of Energy and Environmental Engineering, University of Science and Technology Beijing, Beijing Key Laboratory of Resource-Oriented Treatment of Industrial Pollutants, International Science and Technology Cooperation Base for Environmental and Energy Technology of MOST, Beijing 100083, China
2   Nanjing Academy of Resources and Ecology Sciences, Jiangbeixinqu, Nanjing 211500, China
*   Correspondence: dima.francis@xs.ustb.edu.cn (F.A.F.J.D.); zifuli@ustb.edu.cn (Z.L.);
    Tel./Fax: +86-010-62334378 (Z.L.)

**Abstract:** Organic waste-derived biogas production is an effective way to transform biowaste into renewable energy for the electricity supply in developed and developing countries. This study analyzes the feasibility of biogas production as a solution to waste management and electricity supply in Bangui, the capital city of the Central African Republic. The selection of the biogas plant site in an urban area is a complex process due to the area availability and different factors. The GIS, combined with the MCDA, could analyze the environmental, social, and economic factors and criteria such as slope, settlement, rivers, land, urban growth, and local and major roads. Applying the ELECTRE TRI as the MCDA method enhanced the techniques to determine the suitable biogas plant site. The biowaste amount and distance from the suitable site were determined using the ArcGIS distance toolset. The biogas plant's economic and environmental benefits, such as the electricity production capacity and $CO_2$ reduction, were analyzed based on the population growth and the biogas production per year. The analyzed results obtain an area of 3.5 $km^2$ for a large-scale biogas plant construction, with a potential production of 2,126,799.68 kW per year using combined heat and power and 2,303,100.23 kW by converting the thermal energy to electricity. This large-scale biogas plant could treat 20% of the organic waste per year, cover 60% of the city's electricity demand, and reduce 946,200 kg of $CO_2$ equivalent per year.

**Keywords:** biogas; biowaste; MCDA–GIS; ELECTRE TRI method; Central African Republic

## 1. Introduction

The production of biogas has gradually increased in recent years and has contributed to the promotion of renewable energies and the reduction in pollution [1]. Biowaste, such as food waste, domestic sewage sludge, and industrial wastewater sludge, can be transformed into biogas [2]. The biogas derived from biowaste is a promising practice for the future energy approach [3]. Methanization or biogas production is a natural process for transforming biowaste into energy inside a digester without oxygen; this process is called anaerobic digestion (AD) [1]. The AD techniques can help to solve some urban issues in developed and developing countries, such as the management of urban waste [4], the reduction in greenhouse gases (GHGs) [5], the production of electricity [6], and the production of transportation fuel [7] and natural fertilizer [8].

Bangui, the capital and the largest city of the Central African Republic, has an area of 6700 ha = 67 $km^2$, with 1.5 million inhabitants (Figure 1). Approximately 100,952 households produce around 50,476,000 tonnes of biowaste annually, and the city is facing two significant issues. The first issue concerns the treatment of waste generated by the population of Bangui. Most of the city's biowaste is dumped in public garbage cans or gutters, obstructing

rainwater drainage and causing flooding and pollution. The toilet biowaste is frequently not treated and is abandoned once filled, polluting underground water with its burial. The second issue concerns the current electricity production capacity of 60 MW, which is insignificant for the capital city, causing a lack of electricity in some districts. The demand forecasts estimate a growth in peak demand of 403 MW in 2030. Finding a renewable resource to produce electricity in the city is necessary to fill this electricity gap.

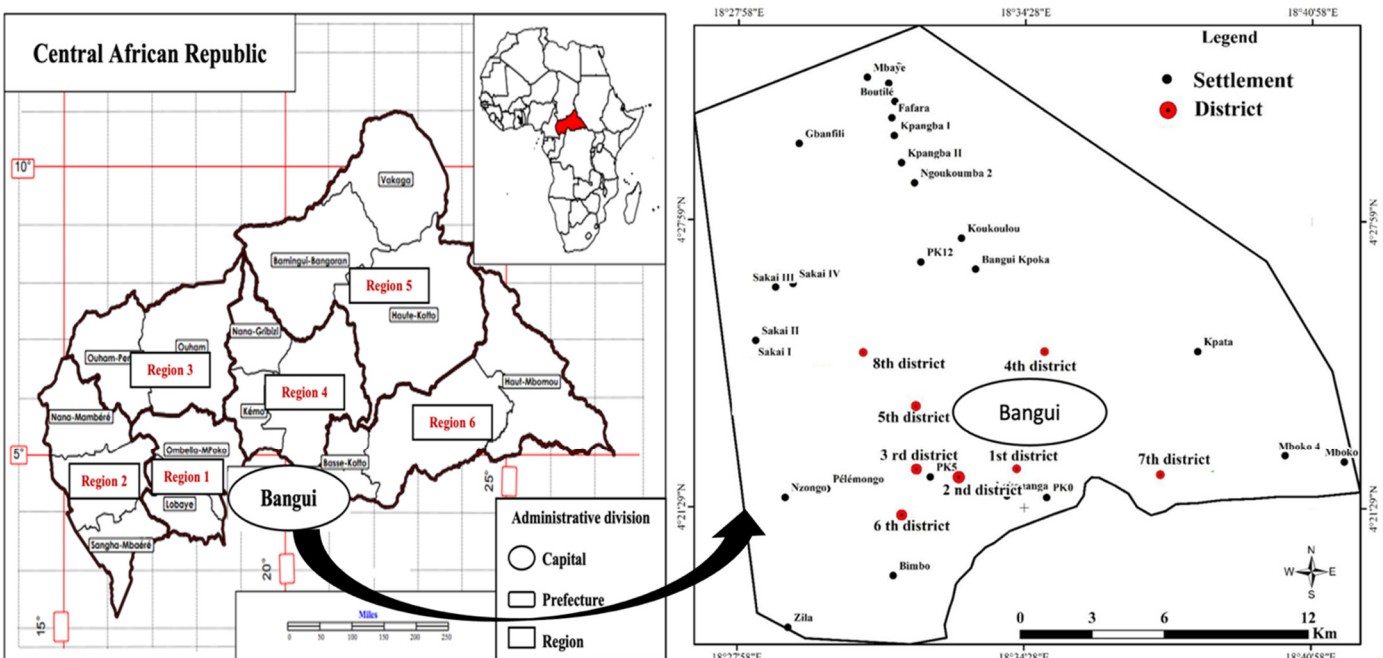

**Figure 1.** Study area.

This research uses methods and techniques to determine a suitable location with respect to the factors and the criteria for biogas plant construction and systematically evaluate its production to solve electricity and waste management problems. Various factors, including biophysical, socioeconomic, and technical considerations, can determine if an area or land is suited for a biogas plant [9]. These methods are based on geographic information systems (GIS) [10], the multicriteria decision aid (MCDA) [11], and the integration of the MCDA–GIS method [12]. The MCDA commonly provides information on locations that allows the evaluation of the criteria from which their choice, ranking, or classification is made in order to manage the decision support system (DSS) [13]. The formal application of the MCDA–GIS can help to resolve spatial decisions [14].

A conceptual framework was created for the criteria analysis [15], with advanced processes, such as the iterative use of the ELECTRE TRI and the outranking method, which provide a novel approach to collecting criteria such as slope, settlement, rivers, land, urban growth, and local and major roads to obtain the suitable location. This study highlights methods that innovate the design phase, such as the iterative application of the ELECTRE TRI to accomplish the MCDA, which defines the evaluated alternatives or options that are contrary to the general process based on the spatial multicriteria decision analysis design [15]. This method can efficiently determine the biogas plant sites in urban areas and estimate the biowaste necessary for the operation, avoiding conflict based on two essential factors: the social opposition by the NIMBY (not in my backyard) consideration and a large amount of social, economic, and environmental data that is necessary for the biogas plant construction.



## 2. Methodology

### 2.1. Estimation of Biowaste

The amount and location of various feedstocks or biowastes, such as domestic waste, were determined using national statistics and population division [16]. The biowaste produced by the population of Bangui was estimated according to the projection of the urban population at three waste generation rates, i.e., 0.5, 0.75, and 1.0 kg/person/day (Table 1).

**Table 1.** Estimation of waste produced by the population.

| Population | Year | 2015 | | 2020 | | 2025 | | 2030 | |
|---|---|---|---|---|---|---|---|---|---|
| | Bangui | 798,000 [a] | | 889,000 [a] | | 1,016,000 [a] | | 1,200,000 [a] | |
| | Generation rate | Tw | Ow | Tw | Ow | Tw | Ow | Tw | Ow |
| **Waste Amount** | 0.50 Kg/person/day | 399,000 | 239,400 | 444,500 | 266,700 | 508,000 | 304,800 | 600,000 | 360,000 |
| | 0.75 Kg/person/day | 598,500 | 359,100 | 666,750 | 400,050 | 762,000 | 457,200 | 900,000 | 540,000 |
| | 1.00 Kg/person/day | 798,000 | 478,800 | 889,000 | 533,400 | 1,016,000 | 609,600 | 1,200,000 | 720,000 |

Tw = total waste, Ow = Organic waste, Unite: tonne/day, [a] = https://populationstat.com/Central-African-Republic/Bangui (accessed on 20 July 2021).

Field research was conducted to collect data on the domestic and industrial biowaste, including the sewage sludge from SODECA, the Central African Republic water distribution company, the food waste from supermarkets (BAMAG and DAMECA), the three most prominent hospitals in the city, and MOCAF, the leading company in the industrial sector, and an agri-food company specializing in the manufacture of beverages. The livestock was categorized by age, gender, and production quantity to estimate the amount of biowaste produced by farm animals. The crop residues consisted of manure, straw, agro-residues (such as greenhouse waste and sugar beet waste), and grass hay and were obtained based on the different crop types (Table 2).

**Table 2.** Estimation of the biowaste by sector of activity.

| Biomass | tTS/yr | tww/yr | Estimated GWh (% of Total) |
|---|---|---|---|
| Municipal biowaste | 8200 | 24,500 | 24 |
| Industrial biowaste | 4000 | 8500 | 11 |
| Municipal WWTP sludge | 15,000 | 70,000 | 28 |
| Manure | 54,300 | 327,400 | 103 |
| Grass silage | 201,414 | 584,630 | 601 |
| Straw | 119,200 | 140,223 | 250 |
| Agricultural waste and side products [1] | 6800 | 47,563 | 24 |
| Total | 408,914 | 1,202,816 | 1041 |

t = tonne, TS = total solids; ww = wet weight, [1] = vegetables, greenhouse waste, potato waste, sugar beet.

### 2.2. Constraints and Criteria

There are no previous studies or regulations on the location or restrictions of biogas plants in Bangui. Indeed, in this study, the first approach was based on the support of experts (specialists in the environmental, geology, agronomy, and GIS field) contributing to the fieldwork. The second approach was based on the existing regulations on similar installations, such as the implementation of the landfill in Bangui or other scientific studies in this same field in Europe (France and Germany) and Asia (China and India).

A conceptual framework was created, including different factors, constraints (Figure 2), and criteria values (Table 3) that should be respected in order to install the biogas plant

safely. The specific innovations, such as the iterative use of the ELECTRE TRI as the MCDA method and the criteria evaluation using GIS, can help to obtain a suitable site for the biogas plant. The site selection process comprises the formal MCDA–GIS interaction. It combines analytical decision methods with GIS functionalities, including zonal statistics, to make a decision matrix that establishes each possibility of the minimum and maximum values.

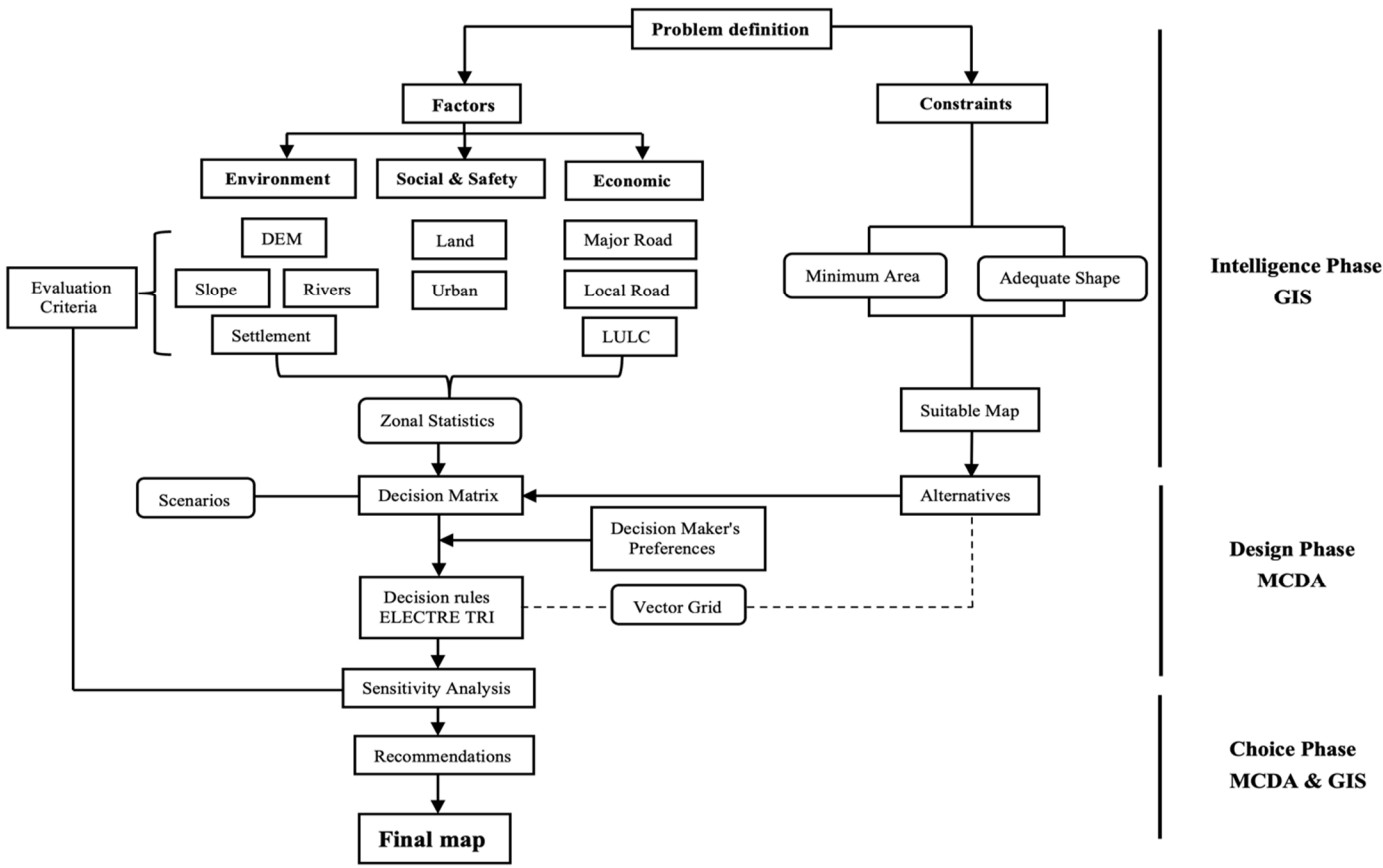

**Figure 2.** The conceptual framework for the biogas plant site.

**Table 3.** Pairwise comparison among the criteria.

|  | Slope | Dem | River | School | Land | Major Road | Local Road | Urban Growth |
|---|---|---|---|---|---|---|---|---|
| Slope | **1** | 1.0 | 0.50 | 0.14 | 0.33 | 0.20 | 1 | 0.11 |
| Dem | 1 | **1** | 0.33 | 0.50 | 0.33 | 0.20 | 3 | 0.11 |
| River | 2 | 3 | **1** | 3 | 4 | 6 | 3 | 7 |
| School | 7 | 2 | 0.33 | **1** | 3 | 5 | 5 | 1 |
| Land | 3 | 3 | 0.25 | 0.33 | **1** | 0.33 | 0.33 | 0.14 |
| Major road | 5 | 5 | 0.16 | 0.20 | 3 | **1** | 1 | 1 |
| Local road | 1 | 0.33 | 0.33 | 0.20 | 3 | 1 | **1** | 0.11 |
| Urban growth | 9 | 9 | 0.14 | 1 | 7 | 1 | 9 | **1** |

Consistency Ratio (CR) = 0.23%.

The fundamental GIS process provides information on the study area and the point shape or suitable site location based on the criteria [17]. The "minimum area" limitation designated the minimal space required for the biogas plant installation. The expertise and investigations allowed us to estimate the biogas plant installation on 1 to 3 km in order to have enough space for stock management and feedstock production through livestock and

agriculture. The biogas plant suitable site map can be made considering the factors, the constraints of each area, and the limitation called "adequate shape" [18].

### 2.3. Outranking Method

The outranking method allows the qualitative and quantitative assessment of the criteria (Table 4), for which preference interval ratios are useless. Therefore, it provides assessment criteria with the various scales to be considered, even if coding them into a single standard scale would be impossible or unusual, thus preventing total compensation across the evaluation criteria and needing less data from the decision maker [19].

**Table 4.** Evaluation of factors defined in the conceptual framework.

| Type | Name | Factors | Objectives |
|---|---|---|---|
| Environmental | C 1 | Distance to the National Agricultural Reserve | Maximize |
| | C 2 | Distance to the river (hydrographic network) | Maximize |
| | C 3 | Occupation and land use (qualitative assessment of adequacy) | Maximize |
| | C 4 | Agricultural soils (qualitative assessment of adequacy) | Maximize |
| Economic | C 5 | Slope (in %) | Minimize |
| | C 6 | Distance to major, national, or local roads. | Maximize |
| | C 7 | Distance to the municipal roads and paths | Minimize |
| Social and safety | C 8 | Distance to school, industrial, commercial, and infrastructure | Maximize |
| | C 9 | Distance to the urban growth (built-up areas) | Maximize |

The multicriteria approach assesses the potential sites (represented by points) with a unique form, size, and uniform area. The MCDA technique requires numerical values; however, providing a unique value of each factor to each point is not practical. We use descriptive statistical metrics and spatialized scenarios to solve this issue and achieve a more specific and thorough suitability rating. We repeat the process on additional smaller sites within these categorized points produced using a vector grid.

#### 2.3.1. Decision Matrix and Spatialized Scenarios

Different values for each alternative were determined using "$M_1$" and "$M_2$" scenarios. $M_1$ represents the highest value, and $M_2$ represents the lowest value. These variables can be categorized when they have the same classification based on the MCDA approach. The classification of alternatives using the decision matrix was based on the values of each factor as determined by the spatial analyst's "Zonal Statistics" ArcGIS tool.

Some sites represented by large points have multiple possibilities, whereas others have only one. Each option needs to be investigated to obtain the homogeneous location and the exact characteristics of the points on all the sites. For this reason, we established a vector grid using ET GeoWizards, a function or extension in ArcGIS that increases geoprocessing functionality, such as data collection, analysis, and topology, in our study. The grid's cell size was assumed to be 3 km, which overlapped with the suitable site, thus converting the large area into small normal point cells and saving the point cells obtained as a vector object.

#### 2.3.2. ELECTRE TRI Method

The ELECTRE TRI method classifies the issues and sorts a set of alternatives into predetermined categories while considering several factors [20]. The ELECTRE TRI classed each option into a category using an outranking approach, creating binary links between each alternative $a_k$ and the $b_i$ and $b_{i-1}$ that bind each category $C_i$. Each profile $b_i$ signifies a suitability category, whereas each $a_k$ represents a suitable site to establish. Therefore, the alternative $a_k$ is assigned to a category or class $C_i$ by comparing each alternative to the reference profiles that define each category's restrictions.

The approach implies the specification of classes in ascending order of preference, with $C_i$ considered the lowest or least suitable for a location; S represents a valued outranking relation that presents $a_k$ as good as $b_i$ [21], thus making an index $\sigma$ $(a_k, b_i) \in [0,1]$. The degree of the authenticity of $a_k S b_i$ is deemed valid if $\sigma$ $(a_k, b_i) \geq \lambda$. With $\lambda \in [0.5,1]$, the lower degree of the validity of ($\sigma$) is lower, thus proving that $a_k$ outranks $b_i$ [22]. The alternative $a_k$ represents the highest category $C_i$, which ranks $b_{i-1}$ (the lower limit) but not $b_i$ (the highest limit) [23].

Decision-Maker's Preferences of ELECTRE TRI

We have three categories of sites: the first category is not suitable (low), the second category is moderately suitable (medium), and the third category is most suitable (high). The ELECTRE TRI classifies and evaluates the sites' suitability based on a category profile ($b_1$ and $b_2$), with $b_1$ defining the limits that separate the moderately suitable and not suitable areas and $b_2$ representing the criteria values that separate the moderately suitable and most suitable sites. (Table 5).

**Table 5.** Reference profiles.

|       | C1   | C2   | C3   | C4   | C5   | C6   | C7   | C8   | C9   |
|-------|------|------|------|------|------|------|------|------|------|
| $b_1$ | 210  | 360  | 4    | 5    | 7    | 260  | 160  | 610  | 610  |
| $b_2$ | 0    | 210  | 3    | 3    | 13   | 160  | 260  | 410  | 410  |
| $q_j$ | 25   | 60   | 0    | 0    | 3    | 30   | 30   | 60   | 60   |
| $p_j$ | 45   | 110  | 2    | 2    | 5    | 60   | 60   | 110  | 110  |
| $v_j$ | 210  | 210  | 3    | 4    | 10   | 160  | 360  | 410  | 410  |
| $k_j$ | 0.20 | 0.20 | 0.10 | 0.20 | 0.20 | 0.10 | 0.10 | 0.20 | 0.20 |

Additional subjective qualities associated with each soft criterion include the weights $k_j$ and the three thresholds: indifference ($q_j$), preference ($p_j$), and veto ($v_j$), which were obtained from the expert participation (Table 5). The veto threshold may show discrepancies or a significant performance gap that prevents an outranking distinct from all other criteria that can accept it. In this work, we set a veto threshold to determine the minimum performance requirements an alternative must meet to be considered for a specific category. The cutoff point was $\lambda = 0:60$, with 60% of the criteria responsible for outranking the criteria weights.

Iterative Application of ELECTRE TRI

We applied ELECTRE TRI twice and used two decision matrices to classify the points, considering the $M_1$ and $M_2$ scenarios defined above. If the classification results of scenarios $M_1$ and $M_2$ are identical, then, in that case, the site is classified in a particular category, with the biogas plant site within this potential site being specified. In contrast, if the classification of the scenarios differs, then a vector grid will be created that overlaps with suitable locations (points) that have not yet been classified. ELECTRE TRI is applied once more to obtain the classification of each grid by using the intersected grid cells as alternatives, considering the same settings for $M_1$ and $M_2$ as the two matrices, in order to obtain the final and most suitable map by determining only the most convenient location.

## 2.4. Determining the Distance to the Biowaste Centers

The distance is measured from each point center to the biowaste center using the ArcGIS distance toolset, which allows Euclidean (straight-line) distance analysis and calculates the direction of each point to determine which biowaste source is the closest to the Euclidean allocation (each most suitable biogas plant).

### 2.5. Biogas Production Capacity on the Site

The cogeneration system can use the combined heat and power (CHP) method, gas engines, or gas turbines to produce electricity. The first solution is appropriate because the gas engines have better electrical outputs for biogas which is rich in methane, as is the case here, and they also present lower investment costs.

2.5.1. Sizing Electricity and Heat Production Capacity by Cogeneration

For an average production of 10,000 t/year of biowaste, or nearly 35 tonnes per day [16], the domestic biowaste can produce 1,000,000 $m^3$/year of biogas, with a biogas yield of 100 $m^3$ per tonne of biowaste. Considering the volume of 1,000,000 $m^3$ produced yearly, with 65% methane, it can obtain 650,000 $m^3$/year. The estimation of the electricity produced per year is shown in Equation (1) [24].

$$\text{Electricity produced [kWh]} = \text{lower calorific value }_{CH4}\text{ [kWh/m}^3\text{]} \times \text{Volume }_{CH4}\text{ [m}^3\text{]} \tag{1}$$

With the lower calorific value = 9.94 kWh/$m^3$ under normal temperature and pressure conditions.

To estimate the recovered electricity produced in one year, we assumed a 5% energy loss to ensure the engine chosen is overfed rather than underfed, as shown in Equation (2) [25].

$$\text{Electricity recovered [kWh]} = \text{Electricity produced [kWh]} \times \text{Engine capacity [\%]}/100 \tag{2}$$

The estimation of the electricity produced per hour is also taken into account and is presented in Equation (3) [25]

$$\text{Electricity}_{y_{1h}}\text{ [kW]} = \text{Electricity recovered [kW]/year} \times 24\text{ h} \tag{3}$$

A motor is designed to operate between 50% and 100% of its rated load, with an optimum efficiency of around 75%. Therefore, we can propose an engine with a power of about 628 kW, close to this optimum [26]. An MG-250 model motor with a capacity of 657 kW, a 72% load, a thermal efficiency of 45.5%, a 38.5% electrical productivity, and 84% of the total productivity of cogeneration with a 16% loss achieves a total production of 253 and 299 kW of electrical power and thermal power, respectively [26]. The annual electricity production estimation is shown in Equation (4) [26].

$$\text{Electricity}_{annual}\text{ [kW]} = \text{Electricity recovered [kW]} \times \text{electrical productivity [\%]}/100 \tag{4}$$

$$\text{Thermal heat [kW]} = \text{Electricity recovered [kW]} \times \text{Thermal efficiency [\%]}/100 \tag{5}$$

To recover the biogas produced by the cogeneration digester, an engine quality MG 250 with the following characteristics should be installed: the power of 657 kW, electric power of 253 kW, thermal power of 299 kW, 84% cogeneration yield, 38.5% electrical productivity, and 45.5% thermal performance and the electricity production of 1654 MWh and the thermal energy of 1955 MWh.

The electricity that can be sold and the heat used are estimated by eliminating the digester's heat and the electrical consumption by the anaerobic digestion units. Therefore, considering the electrical consumption of cogeneration, electrical consumption is estimated to represent 10% of the energy produced. The estimation of electricity consumption by the anaerobic digestion units is shown in Equation (6) [26].

$$\text{Electricity sell }_{annual}\text{ [kW]} = \text{Electricity}_{annual}\text{ [kW]} \times \text{Electrical consumption [\%]}/100 \tag{6}$$

$$\text{Thermal electricity [kW]} = \text{thermal heat [kW]} - \text{Powers [kW]} \tag{7}$$

### 2.5.2. Thermal Heat Converts into Electricity

The heat produced can be used in hot water or hot water circuits for heating digesters and urban buildings or converted into electricity [27]. In our case, neither compost nor WWTP needs heat. Moreover, installing an urban heat system is unnecessary because Bangui has an average temperature of 25 °C to 37 °C, with modest differences during the rainy seasons. Hence, an organic Rankine cycle (ORC) system can be used to convert heat into electricity [26]. The ORC system can start at 50 kWth. Still, in our case, the thermal electricity available is 2,518,579.28 kWth/year, which is equal to 287 kWth/day [26]. We can upgrade to 287 kW thermal heat; so, the turbine is adapted to achieve an average of 5% to 10% heat upgrade to electricity. The electric power of the turbine is shown in Equation (8) [26], and the heat converted to electricity is shown in Equation (9) [26].

$$\text{Turbine power [kW]} = \text{Thermal electricity per day [kW]} \times \text{CEP [\%]}/100 \qquad (8)$$

$$\text{Thermal heat to electricity}_{annual} = \text{Thermal electricity [kW]} \times \text{CEP [\%]}/100 \qquad (9)$$

with conventional electric power (CEP) = 7%.

This result can be added to the annual production and can represent the total of the electricity produced, as shown in Equation (10) [26]

$$\text{Total electricity} = \text{Electricity}_{annual} \text{ [kW]} + \text{Thermal heat to Electricity}_{annual} \text{ [kW]} \qquad (10)$$

## 3. Results and Discussion

### 3.1. The Map Design of Suitable Area for the Biogas Plant

We analyze the eight criteria selected under environmental, social, and economic factors to determine the potentially suitable sites. Due to public concerns about odor and health problems [28], a significant distance between settlements was respected. A minimum distance of 100 m from low land, water wells, and water sources was used, based on the guidelines for biogas plant installation stipulated by the WHO.

The hydrographic network locations less than 150 m from a water source should be removed. The distance from rivers or water bodies should be respected to avoid contamination by the leachate generated by the digester. The points within 70 m of a major, national, or municipal road should be removed. Additionally, for the minimum area, this biogas plant must be installed on land at least 1 km in size. Concerning adequate shape points should be removed ($1 \leq$ area [km] $\leq 1.5$ and compactness $< 0.45$) or ($1.5 <$ area [km] $< 2.5$ and compactness $< 0.25$) (Figure 3).

A 5–10% slope was considered a gentle slope and given the highest score for suitable site selection (Figure 4).

The urban development map was used to select biogas plant locations based on the urban growth. Bangui's land use/land cover (LULC) was created to assess the existing structural plan and provide a vision for the city up to 2030. This projected urban expansion plan was used to conduct a GIS analysis. The process allowed the removal of alternatives located less than 200 m from urban or residential areas (Figure 5).

Based on the criteria, 593 points were obtained in different locations, with 103 points removed because their area was less than 1 km. Therefore, 490 points met the minimum requirement. Twenty-eight points that did not meet the adequate shape constraints were also removed. Thus, 462 remaining points were appropriate (Figure 6).

### 3.2. Determination of Suitable Points as Alternatives

The ELECTRE TRI was used in the scenarios (M₁ and M₂) to classify 462 sites as suitable alternatives. In this case, 160 red points are in the urban area and are classified as not suitable (category 1), 225 yellow points are located at the edge of the metropolitan area but are considered moderately suitable (category 2), and 40 green points are obtained according to the criteria and are considered the most suitable locations for constructing a biogas plant. Twelve not-suitable points alternate with the moderately suitable (categories 1–2);

eleven not-suitable points alternate with the most suitable (categories 1–3); and fourteen moderately suitable points alternate with the most suitable (category 2–3). Therefore, 425 of the 464 points were assigned to the same category, while 37 remained unclassified. The areas were separated into grids to provide a detailed description of the site where the two scenarios do not overlap.

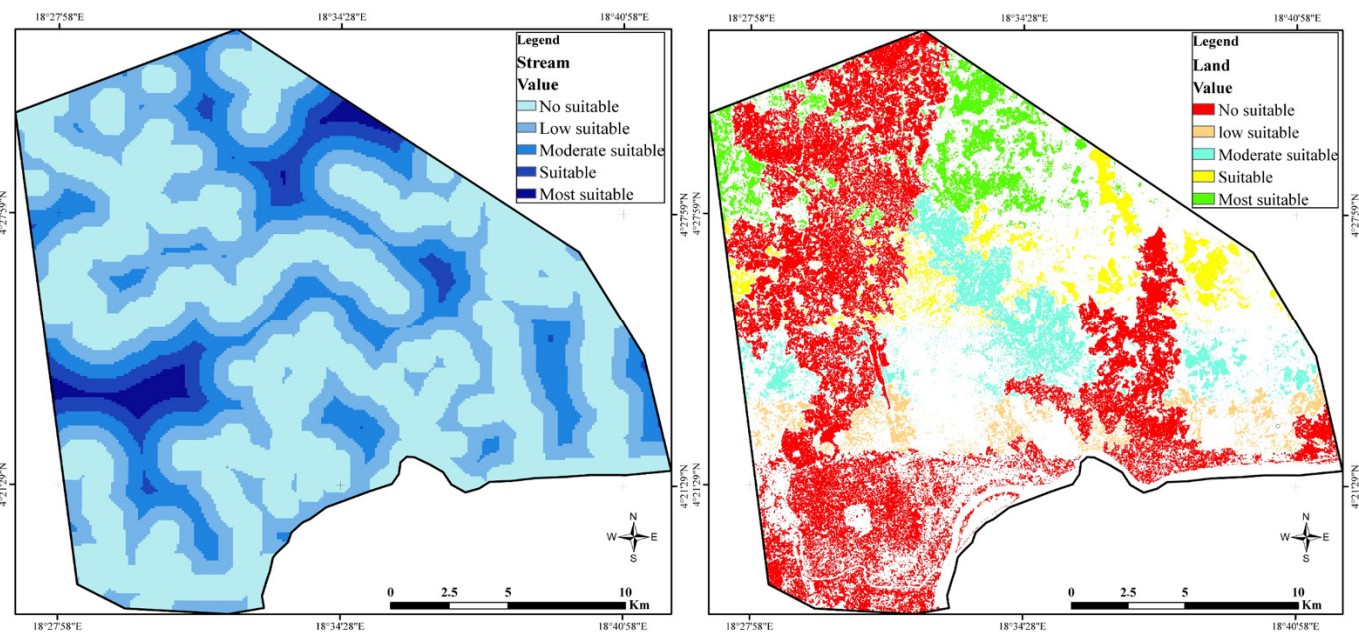

**Figure 3.** Suitable map obtained based on environmental factors and constraints.

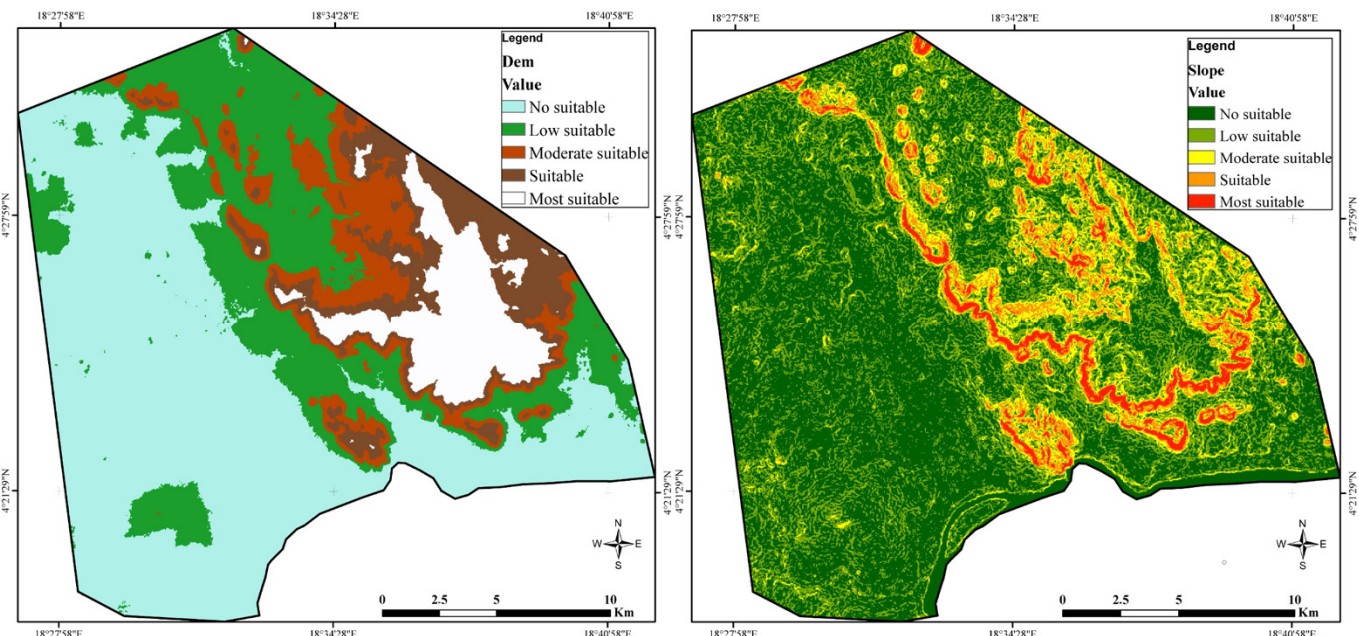

**Figure 4.** Suitable map based on geographical and environmental factors and constraints.

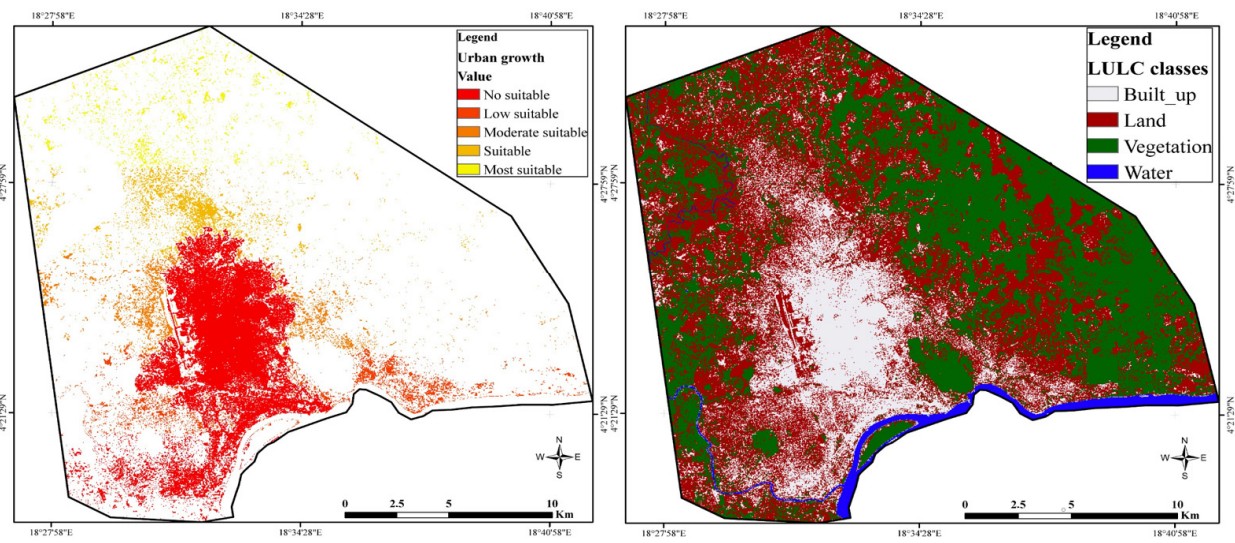

**Figure 5.** Suitable map based on the social factors and constraints.

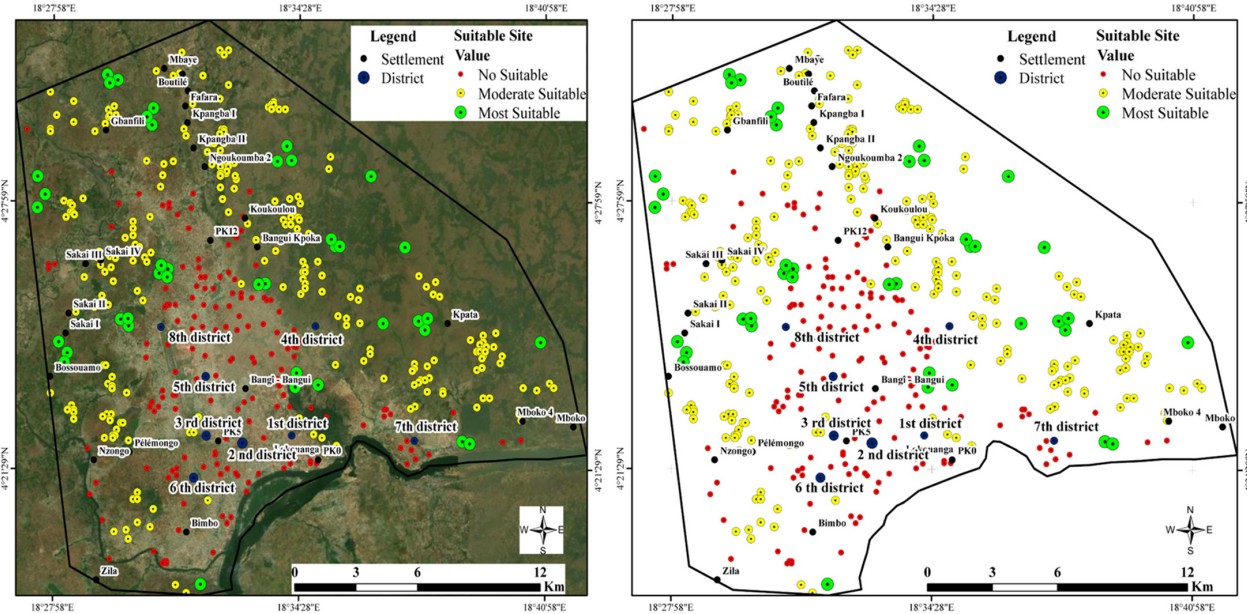

**Figure 6.** Map representing the classification of the suitable and unsuitable points.

### 3.3. Determination of Vector Grids as Alternatives

The vector grid was used in the second phase of the site subdivision to achieve a more detailed classification of the 37 points that were not conclusively classified and had areas ranging from 1 km to 3 km. The ET GeoWizards program created quadrangular grids with 3 hectares in this investigation. However, 505 grids were created by overlapping the quadrangular grid with 37 unclassified sites. Therefore, each grid was required to have a minimum 1 km area. This process removed 376 grids and obtained 129 grids, which were reduced to 106 grids by applying the constraint of compactness $\geq 0.40$. Therefore, 23 grids were removed because of the suitability and shape condition for the biogas plant site.

Considering both cases ($M_1$ and $M_2$), the ELECTRE TRI was utilized, with all 106 vector grids considered options. Thirty-three points were considered as not suitable (category 1); twelve points were considered moderately suitable (category 2); four points were considered as most suitable (category 3); thirty-three not-suitable points alternated with the moderately suitable; six points alternated between not suitable and most suitable; and eighteen moderately suitable points alternated with the most suitable.

Based on the minimum size and adequate shape requirement and the automatically created grids by GeoWizards, some of the 37 potential locations did not have grids. As a result, 15 points were considered moderately suitable and most suitable, and they were included in category 2, based on the $M_2$ scenario, and category 3, based on the $M_1$ scenario, respectively, with 8 grids and 7 suitable points. There were five sites that were the most appropriate and ranged from 1 km to 3 km.

These five sites number from 1 to 5 (Figure 7) are the most suitable locations for the biogas plant construction. They are located in three districts of Bangui, with 55% of the sites in the 8th district, 18% in the 4th district, and 12% in the 7th district, with different geographical coordinates and areas (Table 6).

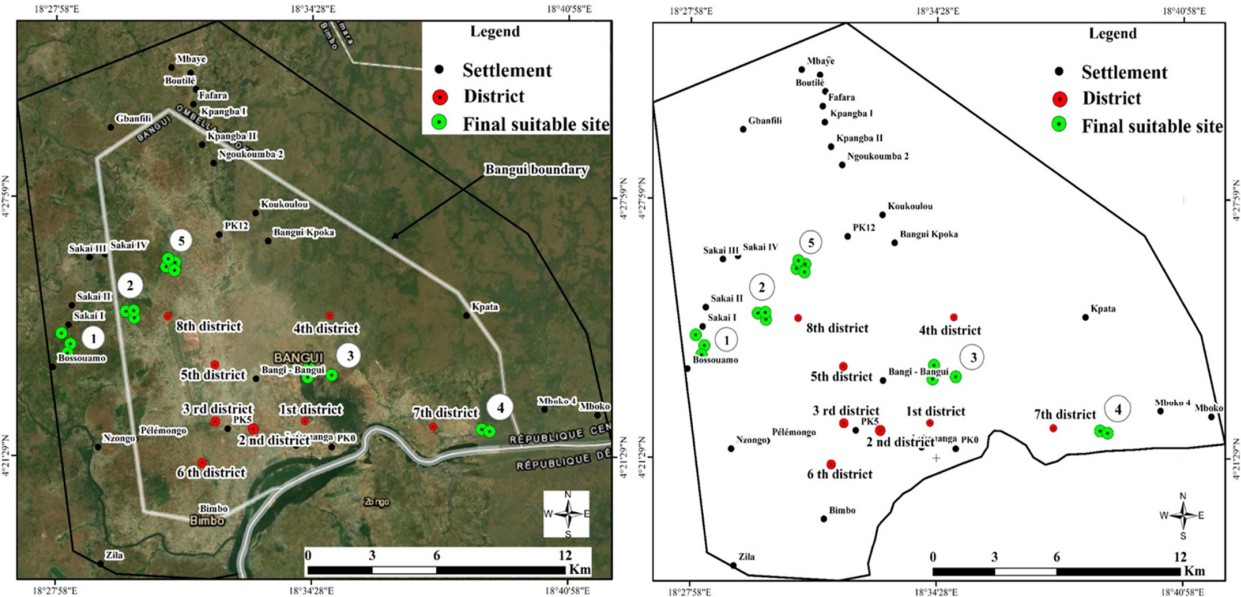

**Figure 7.** Map representing the most suitable sites for biogas plant construction.

**Table 6.** Geographical characteristics of selected sites.

| Locality | X | Y | Area (km$^2$) |
|---|---|---|---|
| **Site (1) Sakai I** | 219,049.284292 | 487,862.563174 | 3.58623 |
|  | 219,468.309417 | 487,372.962217 |  |
|  | 219,329.221672 | 486,949.507225 |  |
| **Site (2) Sakai II** | 222,077.644435 | 488,870.528475 | 2.68432 |
|  | 222,474.520229 | 488,572.87163 |  |
|  | 222,441.447246 | 488,903.601458 |  |
| **Site (3) 4th district** | 230,676.619966 | 486,456.20073 | 2.63325 |
|  | 230,610.474001 | 485,827.814056 |  |
|  | 231,734.955416 | 485,927.033005 |  |
| **Site (2) 7th district** | 238,779.500755 | 483,380.413328 | 2.41951 |
|  | 239,143.303566 | 483,281.19438 |  |
| **Site (5) Sakai IV** | 224,392.753232 | 491,119.491307 | 1.92323 |
|  | 223,962.804455 | 490,921.05341 |  |
|  | 224,062.023404 | 491,284.856221 |  |
|  | 224,359.680249 | 490,755.688496 |  |

*3.4. Biogas Plant Site Optimal Selection-Based Factors*

The site selection was based on the multicriteria decision and the environmental, social, safety, and economic factors mentioned above for implementing the biogas plant. Geographical knowledge of the study area and the contribution of experts in geology, civil engineering, and agronomy helped to select the most favorable sites.

Site number three (4th district), with a surface area of 2.63325 km$^2$, is the closest to the entire biowaste collection center. Still, it is located close to the urban area and a 590 m high slope known as Bas-Oubangui, thus causing the spread of odor, noise, and olfactory pollution. Its location close to the metropolitan area will cause long-term problems in extending the biogas plant or Bangui, considering the city's urban growth.

Site 1 (SAKAI I) was selected for establishing the biogas plant. It has the most suitable and more extensive area (3.58623 km$^2$) among the five most appropriate sites (Table 7) and could manage the biogas plant's operation space and equipment. Moreover, its large area can be used in the case of the extension of the plant or the storage of feedstock or fertilizers from the digesters. It is located far from the urban area, practically situated in the border limit of the urban areas of Bangui, and thus avoids any impact of the plant, such as odor and noise, on the population.

**Table 7.** The distance of the suitable sites to the biowaste collection center.

| Locality | Area (km$^2$) | Distance of Biowaste Centers to the Suitable Sites (km$^2$) | | | | | | |
| --- | --- | --- | --- | --- | --- | --- | --- | --- |
| | | MOCAF | DAMECA | BAMAG | SODECA | General Hosp | Communautaire Hosp | Amitie Hosp |
| Sakai I | 3.58623 | 1,036,391 | 1,226,013 | 1,226,819 | 1,347,573 | 1,193,489 | 9,528,839 | 8,982,043 |
| Sakai II | 2.68432 | 10,929,756 | 10,792,304 | 10,677,917 | 11,716,538 | 10,158,262 | 7,555,027 | 6,452,295 |
| 4th district | 2.63325 | 10,542 | 4187 | 3693 | 3792 | 2751 | 2157 | 2608 |
| 7th district | 2.41951 | 15,811,091 | 858,745 | 8,320,406 | 6,938,459 | 8,327,419 | 10,567,197 | 11,617,289 |
| Sakai IV | 1.92323 | 12,776,391 | 10,913,891 | 10,666,627 | 11,443,348 | 9,956,352 | 7,329,586 | 5,458,734 |

DAMECA: supermarket and food; BAMAG: supermarket; Hosp: hospital.

Site 1 (SAKAI I) is in the same area as sites 2 (SAKAI II) and 4 (SAKAI IV). Still, it is the closest to the road and the agricultural zone, thereby facilitating crop residue (biowaste) collection for the biogas plant. The site 1 is represented by the green square and the biowaste centers are represented by red points and the distance between them is represented by a line (Figure 8).

*3.5. Biogas Production and Valorization of This Site*

3.5.1. Biogas Plant Operation

The biogas plant site is located near a slaughterhouse and agricultural area with a large capacity of feedstock. It can be built on approximately 9000 m$^2$ to 1500 m$^2$ with a digester capacity of 1600 m$^3$ to 1800 m$^3$ of biogas/day. With a current installed capacity of 657 kW, the plant can inject between 6500 and 6600 kWh daily into the electricity network, with possible extensions. A total of 2,303,100.23 kW of electricity can be produced and sold annually (Table 8).

The digestate is subdivided into three products: liquid fertilizer, solid biofertilizer, and organic soil amendment. The operating income is generated by the sale of electricity and digestate in the form of biofertilizers. The revenue generated by the electricity production could be doubled, and a high demand for fertilizers could be observed because of the country's scarcity and the cost of chemical fertilizers.

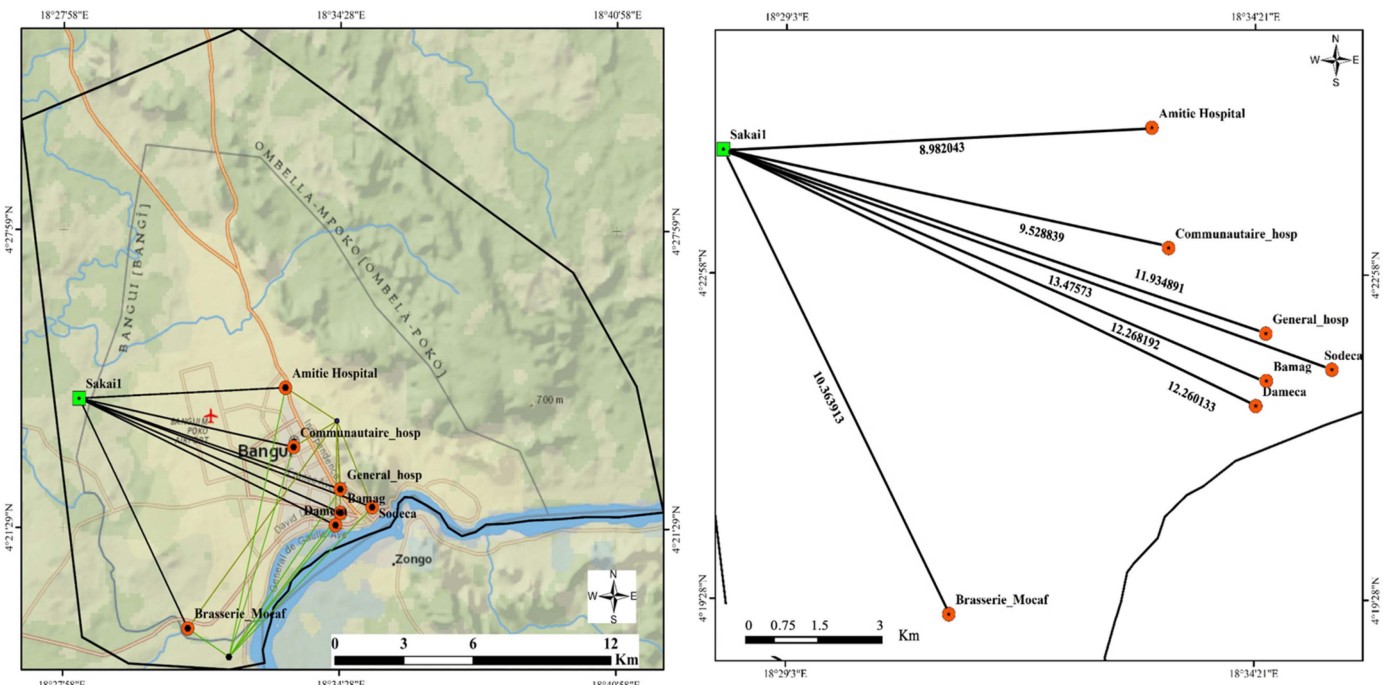

**Figure 8.** Map of the selected biogas plant site and biowaste collection centers.

**Table 8.** Biogas plant production by cogeneration.

| Characteristic | Production |
|---|---|
| **Electricity produced (Equation (1))** | 6,461,000 kWh |
| **Electricity recovered (Equation (2))** | 6,137,950 kWh |
| **Electricity produced in 1 h (Equation (3))** | 700.67 kWh |
| **Electricity $_{annual}$ (Equation (4))** | 2,363,110.75 kWh |
| **Thermal heat $_{annual}$ (Equation (5))** | 2,792,767.25 kWh |
| **Electricity sells $_{annual}$ (Equation (6))** | 2,126,799.68 kWh |
| **Thermal heat $_{annual}$ (Equation (7))** | 2,518,579.25 kWh |
| **ORC turbine power (Equation (8))** | 20.09 kW |
| **Thermal heat to electricity (Equation (9))** | 176,300.548 kW |
| **Sum electricity (Equation (10))** | 2,303,100.23 kWh |

The electricity and emissions saved by biogas cogeneration were evaluated for producing electricity and heat using natural gas [27]. We consider electricity production with fossil resources using a boiler with 50% efficiency to calculate the final energy saved [26]. The electricity production of **8.3** TJ.

Final energy saved = **16.6** TJ

Primary energy saved = **18.26** TJ

### 3.5.2. Environmental Benefits

The environmental benefits of the study are mainly related to the recovery of biowaste from households, toilets, livestock, hospitals, hotels, supermarkets, factories, and slaughterhouses in Bangui to produce electricity and biofertilizer. This process could help to solve waste management and reduce GHG emissions related to waste proliferation by reducing

the use of polluting fossil fuels and enabling the development of renewable energies in the country or throughout the region.

We estimate the avoided emissions if producing 1 GJ of electricity from natural gas emits 57 kg of $CO_2$. With the cogeneration of biogas, the emission of 946,200 kg of $CO_2$ is avoided per year if we transform all the biowaste produced by the population into energy and use it for the electricity supply in Bangui. In that case, the $CO_2$ emissions caused by electricity production using fossil resources could be reduced, reaching 0% emissions in 2030.

### 3.6. Discussion

The study model combines spatial and nonspatial data using the GIS-based method and thus provides extensive understanding and a comprehensive perspective of the biogas plant implementation in urban and rural areas while respecting the environmental, social, and economic factors. This study presents methods that innovate the design phase as the iterative application of the ELECTRE TRI to accomplish the MCDA by defining the alternatives evaluated as contrary to the MC-SDSS process that is developed based on the usual design of the spatial multicriteria decision analysis.

In previous studies, Nas, B. et al. [29] used MCDA–GIS methods in land use assessment. The MCDA method has been associated with GIS to solve a variety of problems, including ecology [30], unfavorable location [31], energy such as solar farm location [32], biogas site location [33], and hybrid renewable energy systems [34], but all were primarily based on the general process for MC-SDSS [15]. In the previous study by Silva, S., et al. [17], the advanced MCDA–GIS and the ELECTRE TRI method used for the biogas plant construction was limited to the resolution of suitable sites; otherwise, it was not based on the estimation of biowaste and the determination of the distance between the biowaste centers and the main biogas production sites.

In this study, we estimated the quantity of biowastes, such as sewage sludge, food waste, livestock, and crop residues produced by the city, in order to ensure the large-scale production and long-term operation of the biogas plant. We geolocated these biowaste sources using the ArcGIS Distance toolset-enabled Euclidean (straight-line) distance to calculate the distance between the suitable sites and the biowaste location. Therefore, this method allowed the selection of a nearby area for the biogas plant construction, considering the distance from the potential biowaste centers in order to reduce the transportation fee of the feedstock. Moreover, the electricity production capacity, the biogas plant site operation system, and the environmental benefits were the determiners that made estimating the economic viability of the biogas plant possible.

The estimation of the volume and collection system of the household and animal biowaste sources (fecal sludge) as feedstock is limited in this study. A collection chain can be set up along the route of the seven potential biowaste centers. However, the strategic selection of biowaste collection centers, their treatment methods by cogeneration, and the conversion of heat into electricity, considering the area's conditions provide a strategic advantage for this biogas plant's operation and economic management.

Based on the quantity of the biowaste that can be recovered and the annual biogas production, the biogas plant was classified according to the NY/T 667 rules for the biogas standard system in China, considering an average of 1300 $m^3$/day; our biogas plant is classified according to NY/T 667–2003 and NY/T 667–2011 as a large biogas plant with daily biogas yields of $Q \geq 300$ and $5000 > Q \geq 500$, respectively [35]. This biogas plant has a larger production capacity than that proposed in the previous studies [29], which have a similar method but do not present production estimates or study the evolution of the plant over time and according to population growth.

This study is more structured on the plan of environmental impact analysis, economic studies, security investigation, and the theoretical and technical planning of the production of biogas than previous studies [36]; the yields and the production of electricity are reason-

able and can be estimated in the future based on the population growth and the biowaste production per year (Figure 9).

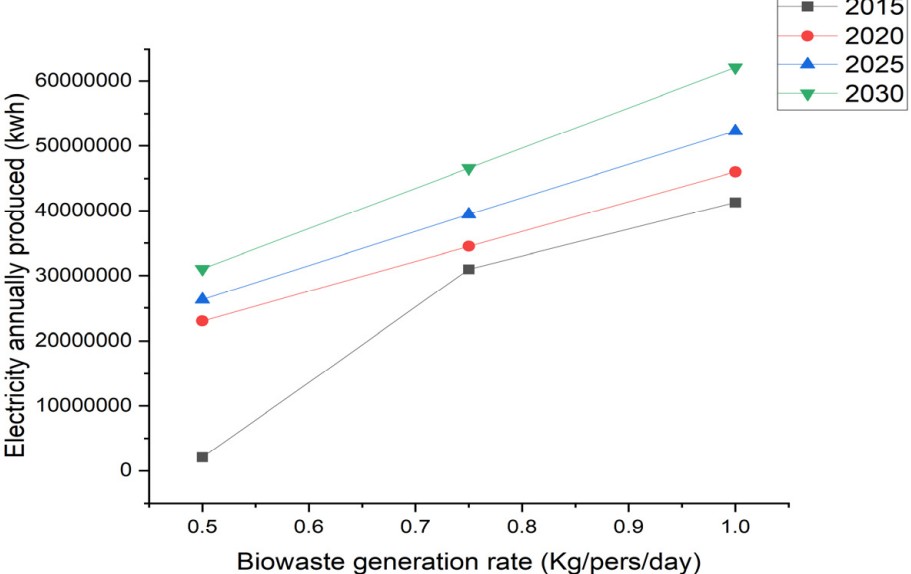

**Figure 9.** Estimation of electricity produced based on the biowaste generated by the population.

## 4. Conclusions

This paper presented the process of valorizing biowaste as renewable energy for implementing urban biogas production; it solved two significant biowaste problems of management and energy supply based on biowaste estimation and the suitable site determination with respect to different urban area criteria, factors, settlement, and regulations. The innovated MCDA–GIS process-based ELECTRE TRI approach examines, classifies, and categorizes alternatives and sites, considering their geographical variability or location, to obtain a suitable area for biogas construction. The production by the cogeneration system proposed in this work was adapted according to the study area's conditions and geographical situation. This production covers 80% of the city's electricity demand for the population, recovering 60% of waste annually and reducing the 946.200 Kg $CO_2$ per year equivalent emissions per year. It can reduce the $CO_2$ emissions caused by electricity production using fossil resources to 0% emissions in 2030, based on the population growth and the biowaste production per year. This study can be used as a model for land localization and biowaste estimation for biogas production in urban areas in developed and developing countries. Therefore, the urban areas, in general, have a challenge concerning the exploitation of the land, and this method makes it possible to locate readily and with more precision the geographical coordinates and the environmental and social conditions of the area to avoid numerous errors.

**Author Contributions:** Conceptualization, F.A.F.J.D. and Z.L.; methodology, F.A.F.J.D.; software, F.A.F.J.D.; validation, Z.L., H.-P.M. and L.Z.; resources, L.Z.; data curation, F.A.F.J.D.; writing—original draft preparation, F.A.F.J.D.; writing—review and editing, Z.L.; visualization, H.-P.M.; supervision, Z.L.; project administration, L.Z.; funding acquisition, Z.L. All authors have read and agreed to the published version of the manuscript.

**Funding:** This research received no external funding.

**Data Availability Statement:** https://populationstat.com/Central-African-Republic/Bangui (accessed on 20 July 2021).

**Acknowledgments:** The authors would like to acknowledge the support of the National Key Research and Development Plan (2019YFC0408700) and the Bill and Melinda Gates Foundation (Research Development and technical assistance to RTTC-China Project (OPP1161151) and Reinvent the Toilet

China Project: Innovative Toilet Solutions and Commercial Activities (OPP1157726)). This study was also supported by the International Science and Technology Cooperation Base for Environment and Energy Technology of MOST.

**Conflicts of Interest:** The authors declare no conflict of interest.

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
