# Peer review of "Feasibility Analysis of Biogas Production by Using GIS and Multicriteria Decision Aid Methods in the Central African Republic"

_sustainability, doi:10.3390/su142013418_

Round 1
Reviewer 1 Report
Comments:
The manuscript entitled “Feasibility analysis of organic waste-derived biogas production for energy supply in Bangui, capital of Central African Republic, by using GIS techniques and multicriteria decision aid methods” by Junior et al details the challenges related to establishing a biogas plant and biogas production in urban area of African continent. The manuscript presents an interesting and important topic related to localization of biogas plants. Authors have attempted a nice study; however, it remains with some problems particularly the language of the manuscript. Following are some of the issues that need to be corrected for possible publication. Therefore, I suggest the authors to revise the manuscript after careful considerations. The below comments will help the authors to make it more appealing and thus improve its quality and overall readability.
Title:
The title of the manuscript is very long and can be revised.
Abstract:
1. Line 15: Introduce the abbreviation at first use, “ELECTRE TRI”.
2. Authors have a scope to add a sentence in the beginning of the abstract for highlighting the research problem related to importance of biogas. Similarly, they can provide a concluding statement for the abstract.
Introduction:
Authors should provide more details related to the need of bioenergy and management of the biowaste in the introduction section. For that they can refer and cite the recent literature.
a) Dar et al. (2021) Microorganisms 9, 1952. https://doi.org/10.3390/microorganisms9091952
b) Kandasamy S, et al (2022) Fuel 308 (2022) 122053. https://doi.org/10.1016/j.fuel.2021.122053
c) Pawar KD, et al (2015) Applied Biochemistry and Biotechnology 175, 1971-1980. http://dx.doi.org/10.1007/s12010-014-1379-z
d) Dar MA, et al (2019) Biocatalysis and Agricultural Biotechnology 20 (2019)101219. https://doi.org/10.1016/j.bcab.2019.101219
Methodology:
1. Line 102: delete the word “of”.
2. In fig. 2, some of the spellings are not correct, e.g., Environmental.
3. Lines 147-148, Authors should write some introduction about the “ET GeoWizards in ArcGIS” software used.
4. Line 186, correct the spelling of, identical.
5. Line 217 & 218, correct the values. 095. Same on lines 203-231 and throughout manuscript.
6. Line 267, what is meant by Electric electricity?
Results
1. Line 291, Delete the repetition (percent).
2. Line 350, correct the phrase, “pants” as “plants”.
3. The title/legends of all the tables and figures 1, 3-7, needs to be elaborated.
Discussion:
1. Line 430, elaborate the statement, “Previous studies…”.
2. The discussion of the study is very short. Authors should elaborate it by providing and comparing the potential performance of the designed biogas plant with some previously used models.
Author Response
The title was revised
Abstract
- ELECTRE TRI was introduced in line 15
- the research problem related to the biogas important was highlighted in abstract
Introduction
The detail on bioenergy and biowaste management was introduced in the Introduction with references.
Methodology
The correction in 1,2,3,4,5,6 part was done.
Result
The correction of point 1,2,3 was done
Discussion
The point 1 and 2 were considered.

Reviewer 2 Report
This study applies a multicriteria approach to identify the site for the installation of a biogas system. The manuscript needs significant improvements. Below are some more specific comments for your consideration.
1. The quality of figures 1, 6, 7, and 8 can be improved. It is hard to read the labels (on the axis and elsewhere). Also, the caption of the figures and table must give the readers enough information to understand the main ideas without reading the text.
2. The objective of the paper is not clearly stated. Most papers end their intro section with a clear statement of the objectives and/or hypothesis to be tested.
3. Please use international units whenever possible (e.g., Mg for weight instead of tons).
4. Estimation of biowaste: Need to justify the rate of daily waste production with results from experimental studies. In the worst case, if such information is unavailable, the authors must conduct their own investigation to estimate the rate to use. The methodology used to estimate the biowaste by sector (table 2) must be described.
5. Section 2.5 has too many unnecessary lines of equations that are defined with constant values. Equations are better when defined with parameters.
6. Line 444 - "In that case, the CO2 emissions caused by electricity production using fossil resources could be reduced, reaching 0% emissions in 2030." Please discuss the implication of reaching 0% fossil fuel by 2030 considering the need to increase electricity generation capacity to serve the entire population.
7. "We estimate the avoided emissions if producing 1 GJ of electricity from natural gas emits 57 442 kg of CO2.". What was the emission intensity of the biogas system?
8. Overall, there is a need to improve the writing style, highlight the scientific merit and novelty of the study, and provide an in-depth explanation and discussion of key findings from this study.
Author Response
Please see the attachment, all the comments and suggestions from points 1 to 8 have been treated and corrected according to the review report. Thanks

Reviewer 3 Report
· 1. Article has been split into several short paragraphs under each section, combine these paragraphs in each section at the possible points to build a flow in reading and understanding.
2. Check the article for typos and language
3. Tables were not mentioned in the main text body.
4. Use the term Table, not Tab.
5. Figures were not cited in the main text
6. In the table, data has been expressed as Kg/pers/day. Use person, not pers.
7. More detailed writing on results and discussion is needed for a better understanding of the results of this study.
8. Figure quality should be improved
9. Advantages and limitations of this study should be clearly added.
10. Conclusion should be supplemented with future perspectives
Author Response
All the comments and suggestions from point 1 to 10 have been treated and correct following the review report. Thanks

Round 2
Reviewer 2 Report
The authors have made significant efforts to address some comments however there are some critical points that are left out.
(1) The captions of the figures and table must give the readers enough information to understand the main ideas without reading the text. The authors stated that "If I give more information, the table and figure will be overloaded and not understandable or if I give more details not necessary the article will be longer", it is hard to agree with when there are so many captions that are only two words ( e.g. Table 4) or similar to other captions within the same study (e.g. Figure 3 and 4).
2. Equations - This is a scientific manuscript for publication and not a class report.
For example, in Line 285, you have this:
Electricity annual = Electricity recovered ×0,385 [26].
= 4,296,565 x 0,385
= 1,654,177.52 kWannual
Critique: (a) The readers know how to do addition and multiplication (the second line of the equation and the third line in unnecessary).
(b) The "0,385" is a parameter value so it must be defined as such in the equation.
(c) confusion between "," and "." when describing a decimal value.
Solutions:
(a) omit lines 2 and 3.
(b) Use parameters to define an equation and report the units.
Electricity annual [kW] = Electricity recovered [kw]× Electrical productivity [%]/100
It should be noted that this exercise shows that the unit does not match the parameters. KW is the unit of power and the unit for electricity is kWh.
Use a table to report the value of the key parameter needed for the equations.
(c) Revise as needed.
The same principle for all the equations needs to be applied.
3. The authors wrote " The process of CHP biogas production is equipped with a system to avoid any emissions and the treatment process is also prepared to minimize the emission so in this section the projection of the emission of the biogas system is impossible but it can be done when the system is operating because all this data will depend on the different parameter of the system." in response to comment 7. The response to this comment raises more questions. (a) What is this system that is avoiding emissions from a CHP system? Is it a carbon capture device? Why it was not deployed for the natural gas scenario as well? If you purposely used assumed it would be deployed only for the biogas system, how the cost was integrated into the overall model? As of now, this is not incorporated into the cost model.
It would be better if carbon footprint values from LCA analysis are used for the environmental benefits calculation.
Author Response
1- The caption of figures and tables was corrected and improved for more understanding
2- All the equations were corrected using parameters to define them and report their units.
3- The answer to the question "What is this system that is avoiding emissions from a CHP system": the CHP exhaust more concentration of methane from incomplete combustion and other gases. So frequent checking, engine settings frequent maintenance, and checking of methane concentrations are necessary to avoid emission, and there is a system of control in the biogas plant to solve this problem. Components containing biogas should be frequently monitored to identify any leaks. This includes investigations with leak detection systems such as methane cameras, portable lasers, and management to avoid releases of Pressure Relief Valves. ​​Thus, closed or gas-tight storage should always be used; avoiding leaks and using closed tanks are the highly recommended and most important ways to reduce the global warming impact of biogas plants. two main sources of odorous pollution come from the production of biogas biomass storage and digestate composting units. Closed-operated hydrothermal hydrolysis has positive effects on the overall control of odors escaping in factories; furthermore, odor emissions occur mainly during high-temperature pretreatments and unclosed.
About the cost question in this research, it was important to present the method of localization of biogas plant and estimation of biowaste that can be used in urban areas of developing countries discuss the cost of this biogas plant or the cost of a system it's not more necessary and it depends on the different country because the price is not the same.

Reviewer 3 Report
The present article entitled "Feasibility analysis of biogas production integrating GIS techniques and multicriteria decision aid methods in the Central African Republic" has been significantly revised by the authors considering the suggested comments. This article can be considered for publication in Sustainability in its present form.
Author Response
My sincere consideration and respect for your decision for this workand we are always available for further corrections or information.

Round 3
Reviewer 2 Report
Line 296 - inconsistency in the reported unit ( 1,606,412.21 kW = 1606.4122 MWh.)
Author Response
In line 296, we convert the result from kW to MWh, but it seems unnecessary, so it can be removed to avoid any misunderstanding.
